# Integrated Transcriptomic and Metabolomic Analysis Revealed Abscisic Acid-Induced Regulation of Monoterpene Biosynthesis in Grape Berries

**DOI:** 10.3390/plants13131862

**Published:** 2024-07-05

**Authors:** Xiangyi Li, Yixuan Yan, Lei Wang, Guanhan Li, Yusen Wu, Ying Zhang, Lurong Xu, Shiping Wang

**Affiliations:** 1Department of Plant Science, School of Agriculture and Biology, Shanghai Jiao Tong University, Shanghai 200240, China; lixiangyi@sjtu.edu.cn (X.L.); viola0214@sjtu.edu.cn (Y.Y.); leiwang2016@sjtu.edu.cn (L.W.); leegh25@sjtu.edu.cn (G.L.); lurongxu@sjtu.edu.cn (L.X.); 2Shandong Academy of Grape, Shandong Academy of Agricultural Sciences, Jinan 250100, China; senwy886@163.com; 3Grape and Wine Institute, Guangxi Academy of Agricultural Sciences, Nanning 530007, China; zhy1974@163.com

**Keywords:** abscisic acid (ABA), monoterpenes, transcriptome, HS-SPME/GC–MS, Ruiduhongyu, grape berries

## Abstract

Monoterpenes are a class of volatile organic compounds that play crucial roles in imparting floral and fruity aromas to Muscat-type grapes. However, our understanding of the regulatory mechanisms underpinning monoterpene biosynthesis in grapes, particularly following abscisic acid (ABA) treatment, remains elusive. This study aimed to explore the impact of exogenous ABA on monoterpene biosynthesis in Ruiduhongyu grape berries by employing Headspace Solid-Phase Micro-Extraction Gas Chromatography–Mass Spectrometry (HS-SPME/GC–MS) analysis and transcriptome sequencing. The results suggested significant differences in total soluble solids (TSS), pH, and total acid content. ABA treatment resulted in a remarkable increase in endogenous ABA levels, with concentrations declining from veraison to ripening stages. ABA treatment notably enhanced monoterpene concentrations, particularly at the E_L37 and E_L38 stages, elevating the overall floral aroma of grape berries. According to the variable gene expression patterns across four developmental stages in response to ABA treatment, the E_L37 stage had the largest number of differential expressed genes (DEGs), which was correlated with a considerable change in free monoterpenes. Furthermore, functional annotation indicated that the DEGs were significantly enriched in primary and secondary metabolic pathways, underlining the relationship between ABA, sugar accumulation, and monoterpene biosynthesis. ABA treatment upregulated key genes involved in the methylerythritol phosphate (MEP) pathway, enhancing carbon allocation and subsequently impacting terpene synthesis. This study also identified transcription factors, including MYB and AP2/ERF families, potentially modulating monoterpene and aroma-related genes. Weighted gene co-expression network analysis (WGCNA) linked ABA-induced gene expression to monoterpene accumulation, highlighting specific modules enriched with genes associated with monoterpene biosynthesis; one of these modules (darkgreen) contained genes highly correlated with most monoterpenes, emphasizing the role of ABA in enhancing grape quality during berry maturation. Together, these findings provide valuable insights into the multifaceted effects of exogenous ABA on monoterpene compounds and grape berry flavor development, offering potential applications in viticulture and enology.

## 1. Introduction

Aroma serves as one of the foremost indicators for assessing the quality of both grapes and grape wines. In grape berries, the occurrence of volatile compounds can be classified into eight distinct groups including monoterpenes, norisoprenoids, aliphatics, higher alcohols, esters, phenylpropanoids, methoxypyrazines, and volatile sulfur compounds [1]. Of these, monoterpene is considered a crucial contributor to the floral and fruity aroma of Muscat grape berries [2,3,4]. For instance, linalool imparts citrus notes, whereas geraniol and citronellol contribute to the aromas reminiscent of roses and sweetness [5,6]. The chemicals present include monoterpenes, norisoprenoids, aliphatics, higher alcohols, esters, phenylpropanoids, methoxypyrazines, and volatile sulfur compounds. These monoterpenes often occur as both free and glycosidically bound molecules in grape berries. The free-type monoterpenes have the direct capacity to enhance the aroma of both grapes and wine. During the fermentation and storage of wine, the glycosidically bound monoterpenes undergo hydrolysis. This process is accelerated with acid and/or enzymes, resulting in the release of volatile aglycones. This has a significant impact on the overall fragrance profile of grapes and wine [7]. Despite the significant advancements in the wine industry, the molecular mechanisms regulating the biosynthesis of glycosidically bound monoterpenoids in grape berries remain poorly understood.

The MEP (2-C-methyl-D-erythritol 4-phosphate) pathway is responsible for monoterpene synthesis in grapevines. At the onset of this pathway, the enzyme Deoxy-D-xylulose 5-phosphate synthase (DXS) initiates the process by converting glyceraldehyde-3-phosphate and pyruvate into 1-deoxy-D-xylulose 5-phosphate (DXP). Through a sequence of six enzymatic steps, DXP undergoes transformation into geranyl pyrophosphate (GPP, C10) within the MEP pathway. Notably, within this pathway, there are at least three enzymes that serve as rate-limiting factors, namely Deoxy-D-xylulose 5-phosphate synthase (DXS), Deoxy-D-xylulose 5-phosphate reducto-isomerase (DXR), and 1-hydroxy-2-methyl-2-butenyl 4-diphosphate reductase (HDR) [8,9]. Previous research has indicated that DXS serves as the primary enzyme that controls the rate of monoterpene production in several plant species [8]. In the case of Arabidopsis and tobacco, the over expression of DXS resulted in a substantial rise in the accumulation level of monoterpenes [10,11]. Moreover, the accumulation of *VvDXS* transcripts shows a strong correlation with the levels of monoterpenes in grapes [12]. Likewise, quantitative trait loci (QTL) analysis unveiled potential causal SNPs found in Muscat varieties, along with distinctive mutations that characterize the Muscat-like aromatic mutants [13]. Furthermore, in vivo investigations in transgenic tobacco plants revealed that the *VvDXS* N284 gene generates an enzyme that exhibits superior catalytic efficiency compared to the *VvDXS* K284 gene [10]. At the veraison stage of grape berries, the express level of *VvHDR* was linked to the concentration of both free and bound monoterpenes [14]. Terpene synthase (TPS) is the last enzyme in monoterpene synthesis [15], and more than 100 TPS have been found in grapes thus far.

Monoterpenes can undergo additional modifications through the actions of other enzymes, including glycosyltransferases (GTs) and cytochrome P450-dependent monooxygenases (CYP450). The terpene glycosyltransferase family is widely distributed and facilitates the conversion of activated sugar donors into free monoterpenes [16,17]. To date, several functions within this family have been discovered, which include VvGT7, VvGT14, VvGT15, and VvGT16 [18,19]. Notably, *VvGT14* has been confirmed to be negatively regulated by VvWRKY40, a transcriptional repressor that binds to the W-box on the *VvGT14* promoter. Moreover, it is of particular significance to note that the expression of VvWRKY40 may exhibit downregulation in response to ABA (abscisic acid). This intriguing regulatory interplay between ABA and VvWRKY40 could shed light on the intricate molecular mechanisms underlying the modulation of terpene glycosyltransferase activity and, subsequently, the biosynthesis of aromatic compounds in grape berries [7].

The biosynthesis of monoterpenes is influenced by a multitude of factors, spanning environmental conditions [14,20,21,22], cultivation practices [23], and the presence of exogenous agents like sucrose and hormones such as methyl jasmonate, ABA, and GA3 [24,25,26]. Moreover, ABA has been reported to initiate veraison of grape berries. Over the recent years, ABA has found extensive application in various plant species, including apples [27], Vitis vinifera, and Arabidopsis thaliana [28]. It has been demonstrated that ABA can improve the quality of grape berries, such as their color, sugar content, and aroma [29,30,31]. Although the role of ABA during anthocyanin biosynthesis has been well described [32], the specific molecular mechanism by which ABA regulates monoterpene accumulation in grape berries still remains unknown.

This study employed a combined analysis of monoterpene metabolites via HS-SPME/GC–MS and the transcriptome using RNA-seq to probe the influence of exogenous ABA on their concentrations, with the overarching goal of enhancing monoterpene accumulation in grape berries. The findings from this work highlight the prospect of optimizing grapevine farming techniques in China, offering the possibility of elevating monoterpene concentrations in berries. These findings carry significant implications for the future enhancement of grape product quality and the cultivation industry in the region and beyond.

## 2. Results and Discussion

### 2.1. Effects of Exogenous ABA on Key Parameters in Grape Berry Development

Berries at E_L35 (berries begin to color and enlarge), E_L36 (berries with in terminate Brix values), E_L37 (berries not quite ripe), and E_L38 were chosen in this research [33], and we selected two distinct concentrations of ABA, referred to as ABA100 and ABA500, for the treatment of grape berries. The physicochemical parameters of both the control and ABA-treated grape samples are visually presented in Appendix A. The total soluble solid content increased gradually during grape ripening. However, throughout development, there was no significant difference between the ABA100 and control groups, while the ABA500 and control groups revealed significant differences across all growth stages. As a result, the ABA500-treated samples were selected for more extensive investigation, revealing a noteworthy disparity in total soluble solids (TSS) between the two treatments at both the veraison and mature stages. Similar to the TSS, ABA500 treatments led to an elevation in pH values. In contrast, exogenous ABA resulted in a reduction in the total acid content in grape berries, distinguishing it from the effects observed on TSS and pH. This finding aligns with our earlier study, demonstrating that ABA has a substantial effect on enhancing the total soluble solids (TSS) of grape berries. As anticipated, following ABA treatment, the concentration of endogenous ABA in grape berries significantly exceeded that of the control group. Notably, the ABA content exhibited a declining trend from the veraison to the ripening stage. These findings are consistent with the observations reported in previous research, where they similarly noted an increase in ABA levels in response to treatment, and a subsequent decrease in ABA content from veraison to ripening in grape berries.

### 2.2. Effects of Exogenous ABA on the Monoterpene Compounds of Grape Berries

The impact of ABA on total free-monoterpene and glycosidically bound monoterpene compounds during grape berries’ development at various stages was investigated. The results showed that the concentration of total free monoterpene increased under both treatments, with a sharp increase appearing at the E_L37 stage under ABA treatment, whereas a significant increase occurred at the E_L38 stage for the control (Figure 1A). There was no significant difference between the two treatments at the E_L35 and E_L36 stages. Similar to the total free-monoterpene level, the total bound-monoterpene content increased during all stages (Figure 1B). Interestingly, the exogenous ABA-treated berries showed a greater concentration of monoterpenes than the control at the E_L36 and E_L38 stages.

Furthermore, cluster heatmap analysis was performed to investigate the influence of ABA treatment on monoterpenes in grape berries. In total, we identified 22 monoterpenes during our investigation (Table 1). The chromatograms of HS-SPME/GC–MS analysis for monoterpenes in grape berries at four development stages are displayed in Appendix A. These free monoterpenes, based on their evolutionary patterns across grape phenological stages, could be categorized into three distinct groups for both treatment conditions. In the first group, free monoterpenes peaked at the E_L37 stage under ABA treatment, followed by a decline in concentration at the mature stage (E_L38). In the control treatment, no significant change was observed (black block in Figure 1C). This group contains four monoterpenes: limonene, terpinen-4-ol, cis-rose oxide, and trans-rose oxide. These compounds mainly contribute to orange and rose flavors in grape berries [34,35]. In the second group, the free-monoterpene concentration increased at the berry maturation stage under ABA treatment, whereas no significant changes occurred in the control treatment (green block in Figure 1C). This group consists of γ-Terpinene, isogeraniol, geranic acid, citral, and citronellol. Prior studies suggested that they mainly contribute to the rose and lemon-like rose flavor of grape berries [36,37,38,39]. Importantly, monoterpene concentrations in the first and second groups treated with ABA were significantly higher than those from controls at the E_L 37 and E_L 38 stages. In the third group, the free monoterpenes rose from the E_L-35 to the E_L 38 stage (red block in Figure 1C). Monoterpenes including linalool, nerol, and geraniol were divided into this group. Previous studies indicated that linalool and geraniol, both prominent monoterpenes, were discovered in Muscat grapes, which play a central role as the principal contributors to the delightful grape-like, citrus, and rose flavors in these grapes [40,41,42]. Consistent with our findings, earlier studies have also shown that the concentration of terpenes in grape leaves and fruits increased following ABA treatment. For instance, D-Limonene levels in leaves and γ-terpinene in fruits exhibited such increases. It was also confirmed that ABA treatment not only leads to an increase in secondary metabolites but also enhances sucrose accumulation [34,43]. Earlier studies have demonstrated that the application of sucrose through spraying can significantly enhance the content of monoterpenes in grapes at both the veraison and maturity stages, which also correlates with our findings [24].

While free monoterpenes exhibited a relatively straightforward evolution pattern, glycosidically bound monoterpenes displayed a more intricate trajectory, as detailed in Table 2. Our cluster analysis identified four distinct groups within this complex evolution pattern. By comparing the evolutionary pattern, the first, second, and third evolution patterns are identical to those found in free monoterpenes. However, the fourth group indicated that ABA treatment led to a decline in bound monoterpenes at the mature stage, while the control treatment resulted in an increase in monoterpenes at the same stage (blue block in Figure 1E). It was found that ABA treatment augmented the monoterpene content in the first three groups. In contrast, within the fourth group, ABA treatment led to an increase in monoterpene concentration at the E_L 35 and E_L 36 stages but resulted in a decrease in concentration during the last two stages (E_L 37 and E_L 38). Glycosidically bound monoterpenes hold significance for wine grapes since, during the winemaking process, these bound monoterpenes can undergo transformation into free monoterpenes, imparting essential flavors to the resulting grape wine [14]. In the case of table grapes, glycosidically bound monoterpenes might be transformed into free forms over the storage period, as the quantity of free monoterpenes was significantly increased at maturation [44].

Furthermore, principal component analysis (PCA) was performed to provide a comprehensive understanding of the monoterpene profile (four developmental stages × two treatments × three biological replicates) [24] (Figure 1D, F). The findings revealed that the composition of monoterpene compounds exhibited significant variations depending on the growth of the berries. There was a clear differentiation seen between different phases of berry growth, mostly impacted by the first principal component (PC1). The contribution rates of PC1, PC2, and PC3 were 63.4%, 12.2%, and 16.2%, respectively. The application of ABA treatment had a significant impact on the composition of free-monoterpene profiles. This resulted in a distinct differentiation between the treated samples and the control samples at the developmental stages E_L37 and E_L38. (Figure 1D). Moreover, significant variations in the profiles of bound-monoterpene compounds were noted between the control (CK) and treated samples at every stage of development (Figure 1F). Finally, following ABA treatment, the concentrations of alcohols and other critical aromatic compounds were also detected to be significantly higher in the E_L37 and E_L38 stages compared to the control group. To provide a comprehensive visual representation of the impact of ABA treatment on grape aroma, we employed a radar map, revealing that ABA treatment has the potential to enhance the overall floral aroma of grape berries during the E_L37 and E_L38 stages. (Figure 2).

### 2.3. ABA-Treated Grape Berries Transcriptome Analysis

The RNA-seq of 24 “Ruiduhongyu” grape berry cDNA libraries yielded 261.81 Gb of clean data. The clean reads from these samples were aligned to the reference genome using available data (https://urgi.versailles.inra.fr/Species/Vitis/Annotations, accessed on 22 June 2022). Number of mapped reads ranged from 64,557,962 in CK-E_L38-1 to 102,695,590 in ABA-E_L35-3, and alignment efficiency was 82.23% to 93.86%. (Appendix A). A total of 977 differentially expressed genes (DEGs) in the ABA-E_L35 vs. CK-E_L35 group were obtained, among which 585 were upregulated and 392 were downregulated (Appendix A). In the ABA-E_L36 vs. CK-E_L36 group, a total of 1210 DEGs were detected, among which 811 were upregulated and 399 were downregulated. Similarly, the ABA-E_L37 vs. CK-E_L37 group showed the presence of 2586 DEGs, among which 970 were upregulated and 1616 were downregulated. The E_L37 stage exhibited the highest number of DEGs, and the PCA of aroma compounds also revealed significant distinctions in free monoterpenes at this stage. These findings suggest that the notable gene differences observed at the E_L37 stage following ABA treatment likely contributed to the variations in free aroma. Furthermore, the ABA-E_L38 vs. CK-E_L38 group showed the presence of 1773 DEGs, among which 861 were upregulated and 912 were downregulated. The volcano plots of these comparisons are presented in Appendix A.

### 2.4. GO and KEGG annotations of DEGs

To gain a more comprehensive insight into the functional annotation of the DEGs, we conducted GO functional enrichment analysis, utilizing all reference genes as the background. The GO enrichment results are presented across three primary categories: biological process (BP), molecular function (MF), and cellular component (CC), as discussed in previous studies [45,46,47] (Figure 3). Most of the DEGs were enriched in the (BP) category, followed by the MF and CC categories. Among the BP category, DEGs were mainly enriched in the immune system process (GO:0002376:), defense responses (GO:0006952), and immune response (GO:0006955). It has been suggested that ABA can stimulate the accumulation of secondary metabolites in plants, including terpenes, thereby enhancing the plant’s immune and defense responses [31]. Several studies suggested that terpenes are directly involved in the immune response of plants [48,49,50]. Therefore, it can be speculated that ABA can enhance the content of terpenes to improve plant immunity.

Additionally, the KEGG enrichment analyses of the DEGs among four experimental groups including ABA-E_L35 and CK-E_L35 (Figure 4A), ABA-E_L36 and CK-E_L36 (Figure 4B), (C) ABA-E_L37 and CK-E_L37 (Figure 4C), and ABA-E_L38 and CK-E_L38 (Figure 4D) were analyzed. The KEGG enrichment analysis findings for all groups were shown in Figure 4. DEGs in all four stages were mainly enriched in the metabolic pathways (ko01100) and secondary metabolic pathways (ko01110), consistent with previous studies [29,51]. Significantly, our findings regarding total soluble solids (TSS) indicate that ABA treatment facilitated sugar accumulation in grape berries. This observation aligns with a previous study that proposed a correlation between high sugar content and the accumulation of monoterpene compounds in grape berries. In Figure 4, it is evident that the monoterpene metabolism pathway (ko00902) displays a significant enrichment in the ABA-treated grapes during the E_L35 stage as compared to the control group. At the E_L35 stage, monoterpenes are primarily synthesized, coinciding with a significant surge in ABA levels, alongside other physiological indicators such as sugar content and aroma concentration. DEGs from the E_L35, E_L36, and E_L37 stages were also enriched in the carotenoid metabolic pathway (ko00906), which fits with a prior investigation [18,30]. The synthesis of monoterpenes takes place through the MEP pathway, whereas the ABA pathway is situated downstream of the carotenoid pathway, itself downstream of the MEP pathway. Our investigation unveiled three pivotal gene expression pathways. It was observed that ABA treatment upregulated genes positioned upstream of the MEP pathway, facilitating the flow of carbon allocation through the MEP pathway. The synthesis of terpenes and ABA is intricately linked to the carotenoid metabolic pathway, which could be attributed to alterations in carbon allocation prompted by ABA treatment.

### 2.5. DEGs Involved in Monoterpene and ABA Biosynthesis in Grape Berries

In order to obtain a deeper understanding of how exogenous ABA affects the genes involved in the manufacture of monoterpene compounds in grape berries, we conducted an expression analysis of these genes, which divided them into five distinct groups (as illustrated with the heatmap in the left portion of Figure 5). Additionally, for a more comprehensive investigation of how ABA treatment affects the accumulation of monoterpene compounds, we conducted K-means cluster analysis of genes associated with monoterpene compound biosynthesis, aiming to uncover distinctive gene expression patterns during grape berries’ development. The R package ‘factoextra’ was used to ascertain the ideal number of clusters, resulting in the generation of five clusters (Figure 5 Right). Cluster 1 exhibited a significant rise in transcript accumulation at E_L-37 followed by a decline at E_L-38 under ABA treatment (Figure 5A). Notably, this cluster comprised several monoterpene synthases, such as linalool synthase *(VIT_00s0385g00020)* and beta-ocimene synthase *(VIT_12s0134g00030)*. In addition, *VvDXS (VIT_05s0020g02130)* and *VvHDR (VIT_03s0063g02030)* were also included in Cluster 1. DXS is the key rate-limiting enzyme in monoterpene biosynthesis in many plant species and the expression of *VvHDR* was associated with the accumulation of both free and bound monoterpenes at the veraison stage of grape berries [14]. It is speculated that the differential expression profiles of these genes are likely related to the concentration of monoterpenes in grape berries. Further, the Cluster 2 genes displayed an increasing trend in expression; however, it is worth noting that the expression levels of these genes were higher in the CK group compared to the ABA treatment group. This Cluster 2 group also included *VvGGPPS (VIT_05s0020g01240)* and *VvZEP (VIT_04s0044g00270)*, both of which are associated with ABA biosynthesis (Figure 5B). The expression level of Cluster 3 reached its maximum at the E_L37 stage and then decreased at the mature stage, respectively (Figure 5C). Cluster 3 encompasses enzyme-encoding genes likely involved in terpene modification, including *VvCYP72A59 (VIT_19s0135g00180)*, *VvCYP72A1 (VIT_19s0135g00230)*, and *VvGT7 (VIT_16s0050g01580)*. Notably, within this group, gene expression from E_L35 to E_L37 was higher in the CK group compared to ABA, while gene expression in the ABA-treated group surpassed that in the CK group at the maturity stage. Cluster 4 genes exhibited a consistent downward trend in expression, with higher gene expression levels in the CK group compared to the ABA treatment group (Figure 5D). Notably, this cluster included genes such as *VvDXR (VIT_17s0000g08390*), *VvPSY (VIT_06s0004g00820)*, and *VvNCED1 (VIT_05s0051g00670)*. Lastly, Cluster 5 exhibited a notable rise in the accumulation of transcripts during the E_L-36 stage, followed by a decline towards maturity (Figure 5E). Genes within this cluster included Myrcene synthase *TPS65 (VIT_00s0271g00030)* and *VvNCED2 (VIT_10s0003g03750)*. Based on the aforementioned findings, it is evident that exogenous ABA exerts a substantial influence on monoterpene compounds, despite the fact that most of the genes associated with their synthesis did not undergo notable alterations in reaction to ABA treatment. This suggests that differential gene expression analysis may not entirely account for the variations observed in the corresponding metabolites.

To provide a clearer representation of the impact of how ABA treatment affects gene expression within the MEP pathway, we have specifically highlighted DEGs related to this pathway, as illustrated in Figure 6. The results reveal significant upregulation of two key upstream genes, DXS and DXR, following ABA treatment. Additionally, ABA treatment led to the significant upregulation of several important genes in this pathway, including VvGGPPS, VvPSY, VvTPS, VvGT14, and VvZEP. Interestingly, ABA treatment showed a suppressive effect on the expression of VvWRKY40 transcription factor. Moreover, the expression level of VvGT14 was notably higher in the ABA-treated group compared to the control group, consistent with prior research findings [7]. Furthermore, ABA treatment exhibited inhibitory effects on VvABA8H, suggesting a potential role in blocking downstream ABA modification and consequently increasing ABA accumulation in grape berries.

### 2.6. WGCNA Analysis of ABA-Induced DEGs Related to the Accumulation of Monoterpenoid Compounds in Grape Berries

In order to comprehend the relationship between ABA-induced gene expression and the buildup of monoterpenoid chemicals in grape berries, weighted gene co-expression network analysis (WGCNA) was conducted. A total of 4428 DEGs between the ABA treated and control groups were selected for WGCNA, of which seventeen modules of DEGs with highly correlated gene expression profiles across samples were identified. The modules exhibited varying gene counts, ranging from 42 to 1235, as detailed in Appendix A, with the dark red module comprising the highest number of genes. We calculated the module eigen gene, depicting the first primary component of gene expression levels throughout every module, to establish connections between consensus modules and different traits. Several of the seventeen modules had significant associations with the quantities of volatile chemicals that were identified, and these modules were distinguished by seven distinct colors (Figure 7A). The analysis of module–trait connections showed that the dark green module consisted of 390 genes that had a strong correlation with the majority of monoterpenes (Figure 7B). In addition, the expression levels of these genes in the ABA treatment group were much higher than those in the control group at the maturity stage, which also corresponded to previous studies. These findings suggested that the application of ABA treatment led to enhanced grape quality during the maturity stage of grape berries.

Within the dark green module, we have discovered a collective of nine genes that are directly associated with the process of monoterpene biosynthesis, with four of them being directly associated with this process. These genes include *VvDXS (VIT_05s0020g02130)*, *VvTPS23 (VIT_12s0134g00090)*, *VvCYP71A1 (VIT_18s0001g13790)*, and *VvCYP71A2 (VIT_02s0025g04850)*. Specifically, VvTPS23 is the enzyme-encoding gene responsible for the synthesis of linalool (Figure 7C). Previous reports have indicated that VvCYP71A2 is predicted to function as a geraniol 8-hydroxylase, and its expression is linked to monoterpene concentrations. Furthermore, it appears that *VvMYBA1 (VIT_02s0033g00410)* has the potential to regulate the expression of *VvCYP71A2* [52]. Furthermore, within the dark green module, we identified eight transcription factors, predominantly originating from the MYB and AP2/ERF transcription factor families. Notably, prior research has indicated that ABA activates VvMYB60, resulting in increased sugar and/or anthocyanin levels in grape berries [53,54]. However, it is worth noting that the AP2/ERF transcription factors identified in the dark green module are typically associated with responses to ethylene regulation, even though grapevines are considered nonclimacteric plants [55]. While ABA is primarily known for its role in regulating color transformation and ripening, it is important to note that there have been reports suggesting that ethylene can influence the maturation and development of grape berries. Additionally, research in grapes has uncovered the involvement of ERF transcription factors in aroma accumulation [56]. Our earlier investigation discovered that five of the ten potential transcription factors screened from the DNA pulldown analysis of the VvGT14 promoter were derived from the AP2/ERF family. Previous reports have also indicated that CitERF71, a AP2/ERF family transcription factor found in citrus and other fruits, can activate the gene CitTPS16, which encodes for terpene synthase. This gene is essential for the formation of E-geraniol in sweet orange fruit [57]. These findings highlight the intricate regulatory network involving ABA, ethylene-responsive AP2/ERF transcription factors, and key aroma-related genes, shedding light on the complex mechanisms underlying grape berry development and volatile accumulation.

### 2.7. Validation of DEGs through qRT–PCR Analysis

In order to assess the dependability of these DEGs generated using transcriptome analysis and gene expression data collected via RNA-Seq, a total of three key genes, including three monoterpene biosynthesis pathway genes (VvDXS, VvTPS, and VvGT14) and three ABA biosynthesis pathway genes (VvZEP, VvNCED, and VvABA2), were selected for qRT–PCR analysis (Figure 8). The findings indicated that the relative expression levels of these genes were in line with the general pattern seen in the deep sequencing data. According to these results, we propose that the RNA-seq data are of high quality, precision, and dependability, making them suitable for conducting informed molecular investigations in the future.

## 3. Materials and Methods

### 3.1. Grape Materials and Growth Conditions

Vitis vinifera ‘Ruiduhongyu’ grape berries were sampled in the Guangxi Academy of Agricultural Sciences double-grape demonstration garden in Nanning, Guangxi Province, China at the geographical coordinates of (22.61° N, 108.24° E) in vintage 2020. North–south oriented rows with a 3.0 m space on a T-shaped trunk were managed in the vineyard. The distance between the two plants was 5.0 m in each row, and the self-rooted vines were planted in 2019. Subsequently, solutions of ABA at concentrations of 100 mg/L and 500 mg/L (referred to as ABA100 and ABA500) were administered, along with 0.05% Tween 20 at seven weeks after blooming (berries were still hard and green, E_L-33 stage) according to a previous study [29,33], with 0.05% Tween solution as the control. Solutions were applied to the berries around sunset to allow them to fully soak in the solution as much as possible while preventing evaporation. This research looked at the possibility of using a randomized block design. Three separate plots were used for each treatment or control, and approximately 500 berries were collected from a total of 20 vines. Berry sampling was performed at four stages (E_L35, E_L36, E_L37, and E_L38 correspond to the stages of berry development where they start to change color and increase in size, have intermediate Brix values, are not fully ripe, and are at the stage of being ready for harvest, respectively. The stages were devised according to the E_L system reported previously [33]. The samples were cryogenically preserved in liquid nitrogen and promptly transported to the laboratory. Prior to extraction, the samples underwent a washing process using distilled water to eliminate any ABA that had not been absorbed from the surface of the berries. Around 30 berries were utilized for the examination of total soluble solids (TSS) and titratable acidity (TA). The leftover berries were cryogenically frozen using liquid nitrogen and kept at a temperature of −80 °C for further examination.

### 3.2. Monoterpene Metabolite Analysis in Grape Berries using HS-SPME/GC–MS

To extract both free and glycosidically bound volatile compounds from the grape berries under investigation, we utilized our previously established extraction method [58]. In brief, we processed around 200 seedless grape berries by first grinding them into a fine powder using liquid nitrogen. Following this, the pulverized grape flesh was mixed with 1 g of polyvinylpyrrolidone (PVPP) and 1 g of glucolactone. Following a 240 min cold maceration at a temperature of 4 °C, the flesh was promptly subjected to centrifugation at a speed of 8000 revolutions per minute at a temperature of 4 °C for a duration of 15 min, resulting in the extraction of clear grape juice. Then, a total of 5 mL of grape juice was mixed with 10 microliters of 4-methyl-2 pentanol (used as an internal standard) and 1 g of NaCl in a 20-milliliter vial covered with a PTFE silicon septum. The vial also contained a magnetic stirrer. The vial was brought to a state of equilibrium at a temperature of 40 °C for a duration of 30 min, with agitation occurring at a speed of 500 rpm. An SPME fiber, specifically a 2-cm DVB/CAR/PDMS 50/30 µm fiber from Supelco in Bellefonte, PA, USA, was subjected to activation at a temperature of 250 °C for a duration of 2 h prior to the extraction of the sample. The SPME fiber, which had been activated, was placed in the headspace of the vial to absorb volatile substances at a temperature of 40 °C for a duration of 30 min, with the same level of agitation. Subsequently, the fiber was put into the GC injector port for a period of 8 min to release the absorbed volatile substances.

Finally, the Cleanert PEP-SPE cartridges (Bonna-agela Technologies, Tianjin, China, 200 mg/6 mL) were preconditioned with 10 mL methanol and 10 mL of Milli-Q water for the extraction of glycosidically bound volatiles. The grape juice (5 mL) was poured into Cleanert PEP-SPE preconditioned cartridges. Water-soluble chemicals were eluted using 5 mL of Milli-Q water, while the cartridge was rinsed with 10 mL of dichloromethane to eliminate the free volatiles. Lastly, the bound volatile precursors were removed from the cartridges with a flow rate of 20 mL of methanol at 2 mL/min. The resulting methanol eluate was evaporated and re-dissolved in 10 mL of citrate-phosphate buffer (2 mol/L, pH 5.0) to dry under a rotary evaporator. Subsequently, an extract was added to 100 μL of the AR2000 solution (100 mg/mL in 2 mol/L citrate-phosphate buffer, pH 5.0) in a 40 °C incubator, which was then sealed and vortexed for 16 h to release the bound volatile aglycons. The released aglycons were also extracted using the same SPME process.

An Agilent 7890N gas chromatography system, equipped with an autosampler system, was utilized in combination with an Agilent 5975C mass spectrometer to analyze volatile chemicals in the grape juice (Agilent Technologies, Santa Clara, CA, USA). In this study, the volatiles were separated on an HP-INNOWAX capillary column with an inner diameter of 0.25 mm and a film thickness of 0.25 m (J&W Scientific, Folsom, CA, USA) under a carrier gas flow rate of 1 mL/min using helium as the carrier gas. The following was the temperature gradient program for the oven: 50 °C (1 min hold), then 3 °C every min to 220 °C, then 5 min at 220 °C. It was decided to keep the ion source temperature at 230 °C and the interface temperature at 280 °C. In this study, full scan mode was used from m/z 30 to 350 and was used throughout. It was necessary to utilize the same chromatographic conditions to determine the retention indices for a C6-C24 n-alkane series for qualitative analysis (Supelco, Bellefonte, PA, USA). To identify volatiles present in our reference standard, we compared their mass spectra and retention indices to the NIST05 library (https://www.nist.gov/srd, accessed on 22 June 2022). Volatiles without a standard were provisionally identified by comparing their mass spectra to both the Standard NIST05 library and retention indices reported in the previous literature [14]. We quantified volatiles with reference standards by measuring the peak area ratio of the standard to an internal standard against a known standard concentration. For volatiles without a reference standard, we quantified them based on a standard with a matching number of carbon atoms or a similar chemical structure as the reference standard. The data acquisition process of the overall experiment is provided in (Appendix A).

### 3.3. RNA Extraction, Llumine Sequencing, and Transcriptome Analysis

We utilized berry samples, comprising both control berries and those treated with ABA500 at the E_L 35, E_L 36, E_L 37 and E_L 38 stages, for total RNA extraction using the Plant Total RNA Isolation Kit (Bioteke Corporation, Beijing, China). An Agilent 2100 bioanalyser (Agilent Technologies, Santa Clara, CA, USA) was used to assess the quality and quantity of the extracted RNA by using a Sigma RT-250 RNA kit (Sigma, St. Louis, MO, USA). In total, 24 libraries were built using Illumina HiSeq X Ten (Illumina Inc., San Diego, CA, USA) to produce 150 bp paired-end reads (each sample consisted of 3 biological replicates). Raw reads were filtered by trimming adaptor sequences, and the clean reads were aligned to the grape reference genome using Tophat v2.0.943 (https://plants.ensembl.org/Vitis_vinifera/Info/Index, accessed on 22 June 2022). The threshold for identifying significant differentially expressed genes (DEGs) was determined by using the absolute value of the log2 (fold change) with fragments per kilobase million (FPKM) §1.

The online application was used to perform Gene Ontology (GO) enrichment and KEGG pathway analysis for the differentially expressed genes (DEGs). (http://cloud.aptbiotech.com/#/product-list, accessed on 22 June 2022). The software package in R 4.3.0 for weighted gene co-expression network analysis (WGCNA) was used to build highly co-expressed gene modules with high-quality genes (Reference). WGCNA network and module detection were conducted using a topological overlap matrix, with a mergeCutHeight of 0.3 and a minimum module size of 30. Associations between modules and features were established with all the genes in each module (the sum of free and bound forms of each monoterpene). Significant modules associated with the traits were combined on the basis of high values of correlation (>0.8) and *p* < 0.05.

### 3.4. Validation of Transcriptome Data Using qRT–PCR Analysis

DNase-treated RNA was transcribed with a cDNA synthesis kit to confirm the accuracy of the transcriptome analysis (TransGen Biotech, Beijing, China). The qRT–PCR iQ5 Connect real-time system was operated on a ChamQ SYBR qPCR Master Mix (Vazyme#Q311, Nanjing, China) (Bio-Rad, Hercules, CA, USA). The kit’s instructions included 41 qRT–PCR operations, comprising 95 °C pre-denaturation for 30 s, 95 °C denaturation for 10 s, and 30 s rinsing for a total of 40 cycles at 60 °C. The relative gene expression level was evaluated using a QuantStudio 3D thermocycler from three separate biological replicates using qRT–PCR (Bio-Rad, Hercules, CA, USA). The primer sequences are presented in Appendix A. *VvActin* (EC969944) and *VvGAPDH* (CB975242) were used as internal reference genes, and their relative expression levels were normalized using the 2^−ΔΔCt^ method [59].

### 3.5. Extraction and Determination of ABA

The retrieval of ABA was conducted in accordance with our previous study. We pulverized seedless grape berries into a powder using liquid nitrogen and then mixed 50 mg of the powder with an extraction solution composed of 80% methanol (*v*/*v*). The supernatant, obtained after centrifugation at 10,000 rpm for 20 min, was passed through a SepPak C18 cartridge (Waters, Milford, MA, USA) to eliminate polar compounds and was analyzed in triplicate. The elution process was employed to quantify the ABA content utilizing a UPLC-HRMS system. A C18 column (Waters, Milford, MA, USA) with dimensions of 3 µm × 4.6 mm × 100 mm was utilized to separate the compounds. The elution solution was composed of solvent A, which contained 0.1% acetic acid in water, and solvent B, which included 0.05% acetic acid in acetonitrile. The flow rate was always set at 0.3 mL/min, and the gradient elution program followed the following sequence: from 0 to 6.25 min, the concentration of B was 10%; from 6.25 to 7.5 min, the concentration of B was 40%; from 7.5 to 10.6 min, the concentration of B was 90%; and from 10.6 to 13.5 min, the concentration of B returned to 10%.

### 3.6. Statistical Analysis

The data underwent statistical analysis via one-way ANOVA and Duncan’s multiple range test, utilizing SPSS version 19.0 (IBM Corp., Armonk, NY, USA), with significance set at *p* ≤ 0.05. Heatmaps illustrating metabolite concentrations were generated using R-3.6.1 software, incorporating the pheatmap package. The factoextra, ggplot2, and scatterplot3d tools in R-3.6.1 were used to make three-dimensional loading plots that show the principal component analysis (PCA) of the chemical profiles. Pearson’s correlation value and a two-tailed test were used to conduct correlation studies.

## 4. Conclusions

This research examined the impact of exogenous ABA treatment on the growth of grape berries and the accumulation of monoterpene substances. ABA treatment, particularly at the ABA500 concentration, significantly influenced various physicochemical parameters, including total soluble solids, pH, and total acid content, during grape ripening. Moreover, ABA treatment led to a notable increase in endogenous ABA levels in grape berries, suggesting a positive feedback loop. Monoterpene compounds, crucial for grape aroma, exhibited distinct patterns under ABA treatment, with a significant increase in specific compounds at different developmental stages. The study of the transcriptome identified a large number of genes that were expressed differently and were linked to the pathways of ABA and monoterpene production. The use of WGCNA allowed for the identification of crucial DEGs that are associated with the accumulation of monoterpene compounds, further illuminating the intricate regulatory network involving ABA, transcription factors, and aroma-related genes in grape berry development. These findings provide valuable insights into the mechanisms by which ABA influences grape quality and aroma, with implications for viticulture practices and grape-based product development.

## Figures and Tables

**Figure 1 plants-13-01862-f001:**
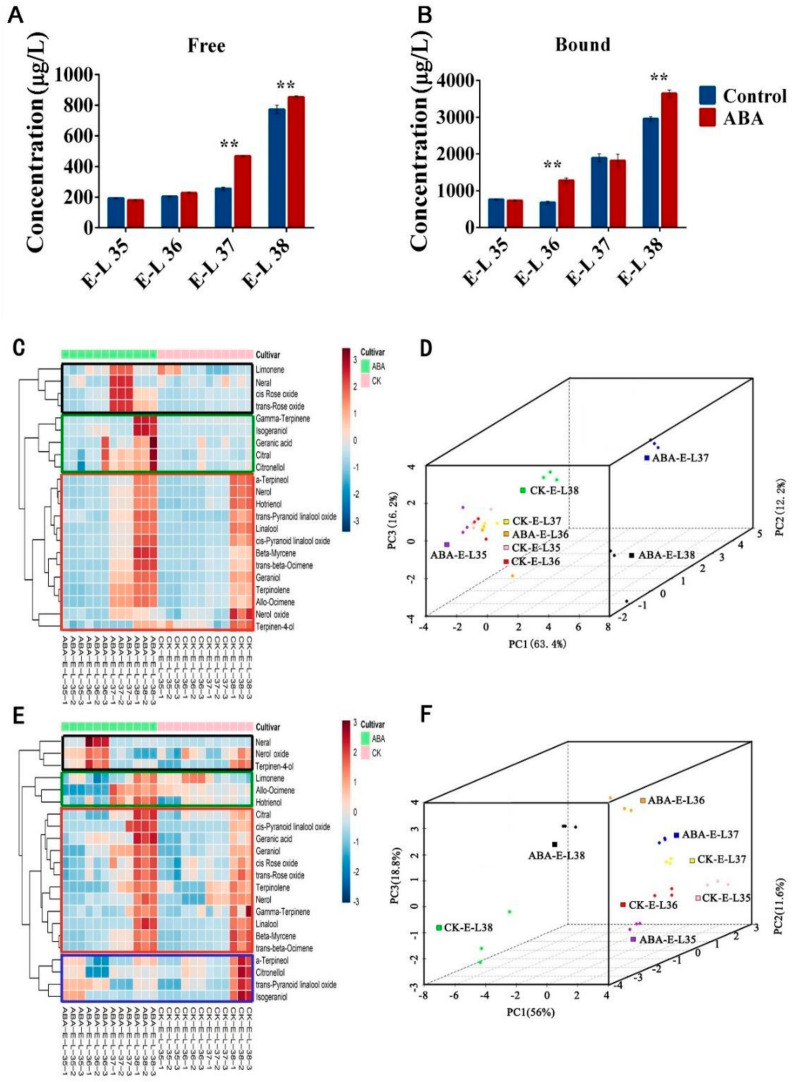
Influence of ABA treatment on the concentration of free and bound monoterpenes. Cluster I is represented by the green block, Cluster II is represented by the black block, Cluster III is represented by the red block, and Cluster IV is represented by the blue block. (**A**) Changes in the total free-monoterpene concentrations during grape berry maturation. (**B**) Modifications of the total glycosidically bound monoterpene concentrations under ABA treatment and control along with grape berry maturation. (**C**) Free monoterpenes in the berries under ABA treatment and control treatment. (**D**) PCA of free monoterpenes. (**E**) Glycosidically bound monoterpenes in the berries under ABA treatment and control. (**F**) PCA of bound monoterpenes. ** indicates significant differences (Duncan’s multiple range test, *p* ≤ 0.01).

**Figure 2 plants-13-01862-f002:**
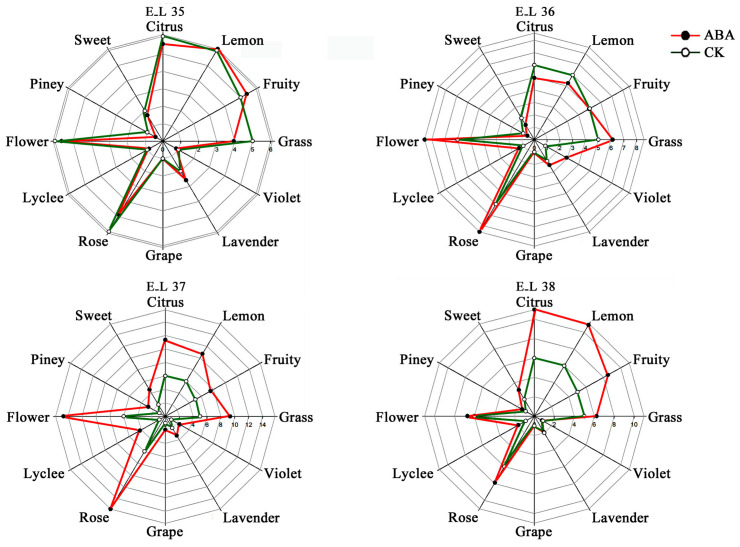
Aromatic series values of grape berries for the different developmental stages under ABA treatment and control treatment (CK) (*n* = 3).

**Figure 3 plants-13-01862-f003:**
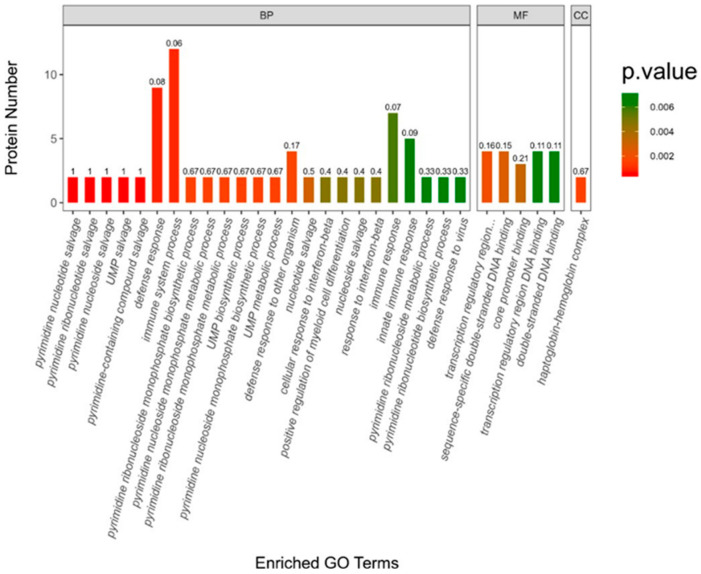
Enrichment analysis was used to determine the number and gene ontology (GO) categorization of differentially expressed genes (DEGs) in various comparisons between ABA treatment and CK. The differentially expressed genes (DEGs) were categorized into three groups based on their involvement in biological processes (BP), molecular functions (MF), and cellular components (CC).

**Figure 4 plants-13-01862-f004:**
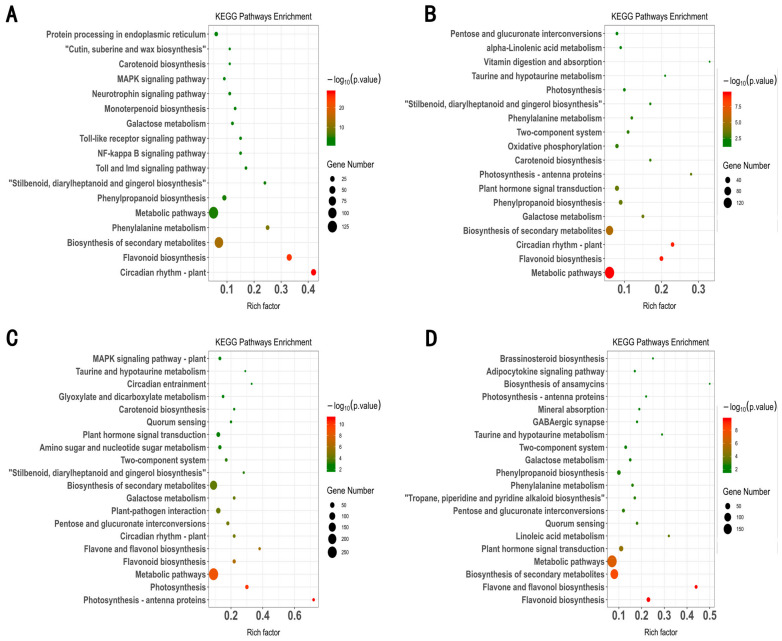
KEGG enrichment analysis of the DEGs between (**A**) ABA-E_L35 and CK-E_L35, (**B**) ABA-E_L36 and CK-E_L36, (**C**) ABA-E_L37 and CK-E_L37, and (**D**) ABA-E_L38 and CK-E_L38.

**Figure 5 plants-13-01862-f005:**
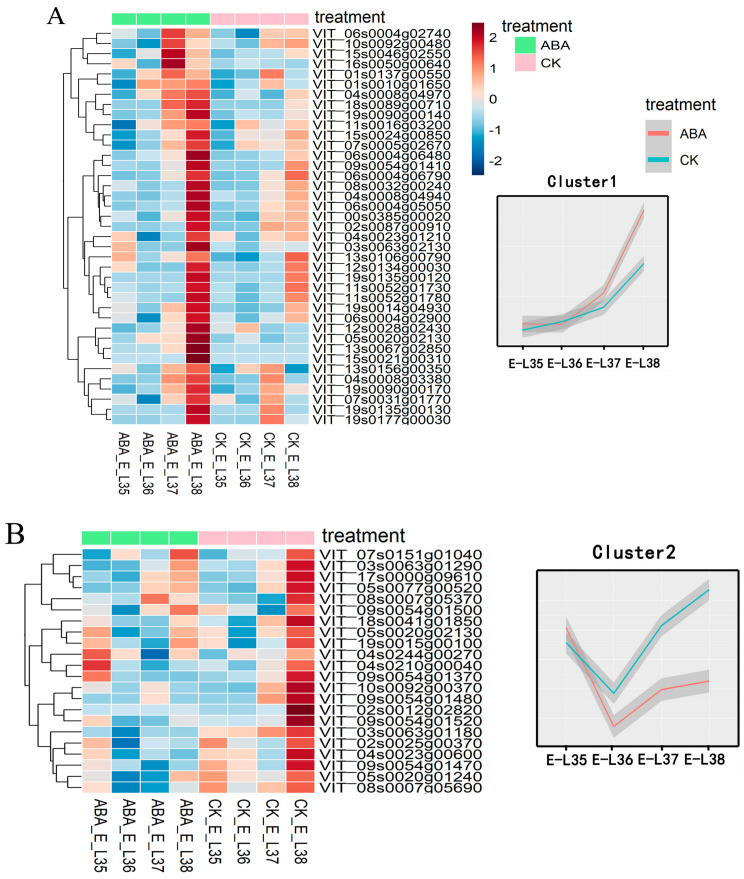
Heatmap (**Left**) and k-means (**Right**) cluster analysis of the time series for genes associated with the biosynthesis of terpene compounds. (**A**) Cluster 1; (**B**) Cluster 2; (**C**) Cluster 3; (**D**) Cluster 4; and (**E**) Cluster 5.

**Figure 6 plants-13-01862-f006:**
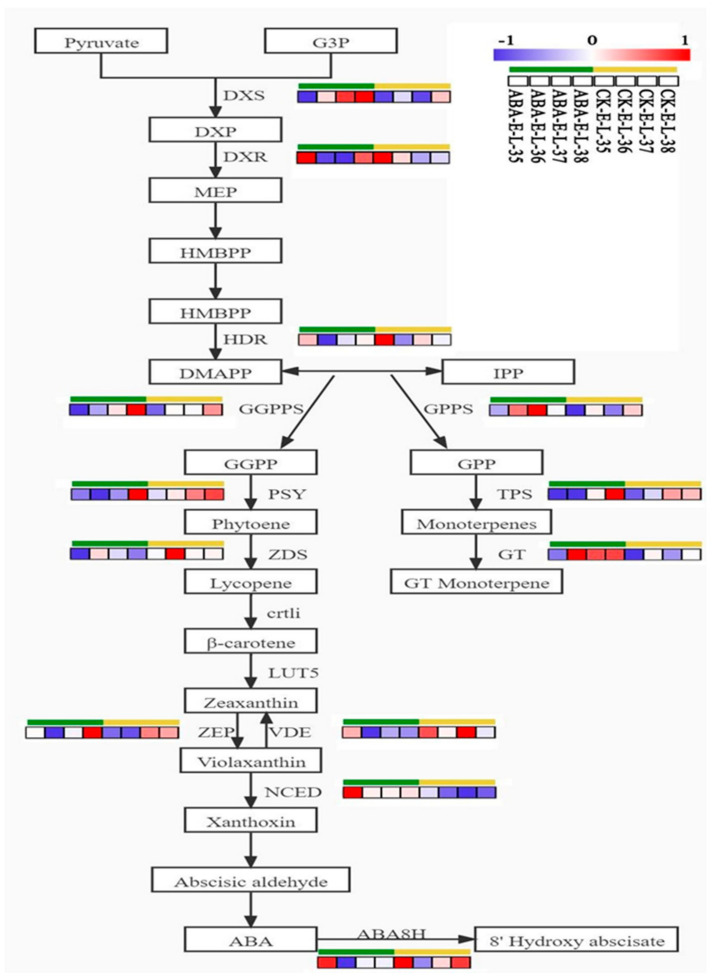
Alterations in the expression of crucial genes associated with the production pathways of monoterpenes and abscisic acid (ABA) in grape berries. The heatmaps were used to assess the amount of transcript abundance by scoring the log2-transformed fold-change values for each experimental group.

**Figure 7 plants-13-01862-f007:**
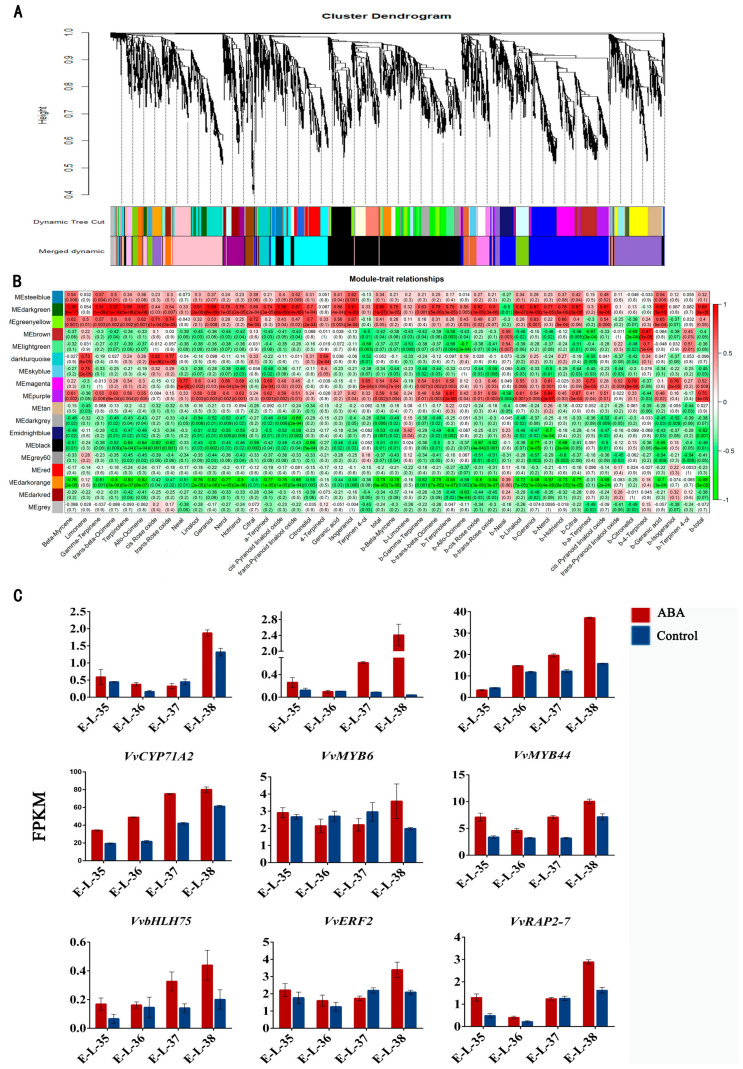
Weighted gene co-expression network analysis (WGCNA) of genes differentially expressed due to ABA, and the hierarchical clustering of genes linked to the synthesis of monoterpenes. (**A**) Hierarchical cluster tree showing 17 merged modules of co-expressed genes. (**B**) Correlations between modules and traits, along with associated *p*-values, are depicted; the left panel presents the 17 modules, while the right panel illustrates a color gradient representing module–trait correlations ranging from −1 to 1. (**C**) The FPKM values of genes involved in monoterpene biosynthesis within the dark green module, compared between ABA treatment and control (CK).

**Figure 8 plants-13-01862-f008:**
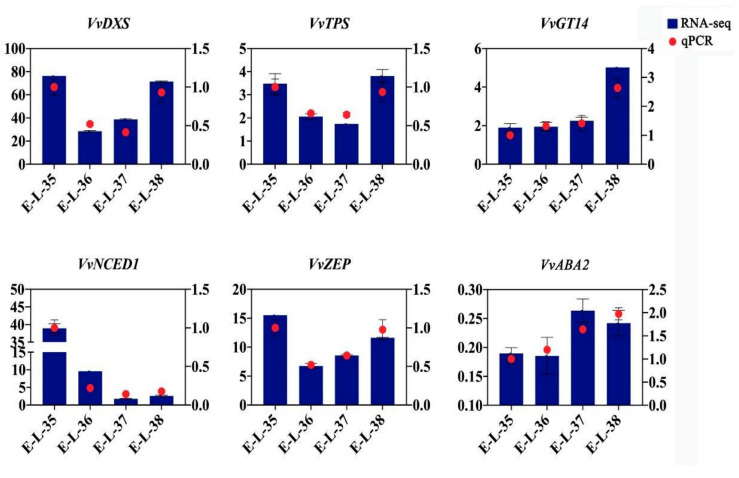
FPKM of RNA-Seq (left y-axis) and the data received from qRT–PCR (right y-axis) for the genes implicated in the monoterpene and ABA biosynthesis pathway.

**Table 1 plants-13-01862-t001:** Changes in free monoterpenes in grape berries at four developmental stages.

	Retention Time	Retention Index	ABA-E_L35	ABA-E_L36	ABA-E_L37	ABA-E_L38	CK-E_L35	CK-E_L36	CK-E_L37	CK-E_L38
Beta-Myrcene	13.32	1164	14.04 ± 0.17	15.58 ± 0.47	24.66 ± 0.1	52.63 ± 1.04	14.8 ± 0.09	15.01 ± 0.16	17.95 ± 0.45	28.69 ± 1.12
Limonene	14.65	1193	36.52 ± 1.92	44.34 ± 1.91	61.5 ± 0.25	42.3 ± 0.64	55.1 ± 2.87	38.39 ± 1	31.64 ± 0.23	38.54 ± 0.15
trans-beta-Ocimene	16.25	1240	0.31 ± 0.08	0.2 ± 0.01	3.94 ± 0.02	9.21 ± 0.02	0.45 ± 0.03	0.62 ± 0.29	1.52 ± 0.17	4.4 ± 0.25
Gamma-Terpinene	17.25	1130	1.93 ± 0.24	2.42 ± 0.15	1.69 ± 0.01	16.83 ± 0.29	1.94 ± 0.03	3.25 ± 0.03	3.36 ± 0.25	3.38 ± 0.06
Terpinolene	18.31	1290	0.34 ± 0.13	1.04 ± 0.07	6.25 ± 0.03	7.59 ± 0.14	0.73 ± 0.13	1.65 ± 0.1	2.2 ± 0.23	5.36 ± 0.11
cis Rose oxide	21.10	1356	114 ± 0.05	2.31 ± 0.16	8.8 ± 0.04	3.91 ± 0.14	1.29 ± 0.07	1.67 ± 0.03	2.08 ± 0.16	2.11 ± 0.15
Allo-Ocimene	21.52	1366	0.55 ± 0.08	1.31 ± 0.09	7.91 ± 0.03	9.5 ± 0.34	0.91 ± 0.16	2.08 ± 0.13	2.79 ± 0.29	6.27 ± 0.59
trans-Rose oxide	21.70	1370	0.47 ± 0.02	0.69 ± 0.03	2.49 ± 0.01	1.4 ± 0.05	0.5 ± 0.01	0.58 ± 0.01	0.68 ± 0.03	0.77 ± 0.06
Nerol oxide	26.05	1474	2.72 ± 0.19	2.28 ± 0.05	4.91 ± 0.02	4.06 ± 0.37	2.36 ± 0.71	3.36 ± 0.14	2.73 ± 0.47	7.74 ± 0.73
Linalool	29.05	1547	43.56 ± 0.51	51.93 ± 1.15	126.9 ± 0.52	292.15 ± 0.64	42.41 ± 0.28	50.1 ± 0.42	64.33 ± 2.52	286.52 ± 10.88
Hotrienol	31.64	1613	11.58 ± 0.28	15.46 ± 0.83	27.46 ± 0.11	51.39 ± 0.82	7.41 ± 0.92	11.69 ± 0.19	20.86 ± 2.07	57.77 ± 1.66
Terpinen-4-ol	31.78	1603	1.34 ± 0.11	1.39 ± 0.1	1.33 ± 0.01	1.91 ± 0.01	1.8 ± 0.15	1.66 ± 0.02	1.41 ± 0.14	2.19 ± 0.05
Neral	34.52	1689	0.09 ± 0.01	0.07 ± 0	0.18 ± 0	0.07 ± 0	0.08 ± 0.01	0.08 ± 0	0.09 ± 0.02	0.09 ± 0.01
a-Terpineol	35.02	1703	3.62 ± 0.14	5.76 ± 0.29	9.96 ± 0.04	26.59 ± 1.18	3.13 ± 0.17	4.95 ± 0.04	8.17 ± 0.64	29.75 ± 0.79
Citral	36.42	1741	31.54 ± 0.18	32.4 ± 0.79	32.47 ± 0.13	33.25 ± 0.75	31.74 ± 0.02	31.77 ± 0.28	31.89 ± 0.02	31.79 ± 0.16
cis-Pyranoid linalool oxide	36.45	1742	3.19 ± 0.13	7.46 ± 0.66	22.52 ± 0.09	75.08 ± 0.71	2.42 ± 0.03	4.04 ± 0.05	8.12 ± 0.47	56.1 ± 4.67
trans-Pyranoid linalool oxide	37.27	1765	3.05 ± 0.02	2.24 ± 0.13	4.39 ± 0.02	7.34 ± 0.05	2.39 ± 0.04	2 ± 0.03	2.09 ± 0.08	6.14 ± 0.34
Citronellol	37.47	1770	6.81 ± 0	7.02 ± 0.17	7.08 ± 0.03	7.25 ± 0.15	6.9 ± 0	6.94 ± 0.07	6.89 ± 0.01	6.96 ± 0.03
Nerol	38.81	1808	4.27 ± 1.13	12.04 ± 0.65	46.85 ± 0.19	102.42 ± 2.19	4.96 ± 1.19	4.13 ± 0.07	19.98 ± 1.2	115.06 ± 3.11
Isogeraniol	39.10	1816	0.12 ± 5.53	0.81 ± 0.15	0.12 ± 0	4.2 ± 0.2	0.11 ± 0	0.12 ± 0	0.12 ± 0	0.34 ± 0
Geraniol	40.45	1855	12.06 ± 0.21	20.46 ± 1.16	66.57 ± 0.27	103.92 ± 1.12	11.58 ± 2.28	19.87 ± 0.18	26.14 ± 1.31	81.88 ± 3.33
Geranic acid	56.13	2350	0.9 ± 0.11	0.91 ± 0.02	0.9 ± 0	0.94 ± 0.02	0.9 ± 0	0.9 ± 0.01	0.9 ± 0.01	0.9 ± 0

**Table 2 plants-13-01862-t002:** Changes in bound monoterpenes in grape berries at four developmental stages.

Monoterpenes Profiles (μg/L)	Retention Time	Retention Index	ABA-E_L35	ABA-E_L36	ABA-E_L37	ABA-E_L38	CK-E_L35	CK-E_L36	CK-E_L37	CK-E_L38
Beta-Myrcene	13.32	1164	21.65 ± 0.58	29.84 ± 1.38	45.56 ± 0.76	74.41 ± 2.99	20.23 ± 0.64	28.51 ± 0.7	31.28 ± 1.11	81.94 ± 5.28
Limonene	14.65	1193	73.81 ± 9.53	54.68 ± 7.3	69.44 ± 5.3	97.41 ± 3.37	79.23 ± 1.79	97.89 ± 1.34	74.86 ± 7.2	64.96 ± 3.37
trans-beta-Ocimene	16.25	1240	1.48 ± 0.03	2.87 ± 0.38	5.36 ± 0.21	11.76 ± 0.56	1.28 ± 0.1	2.52 ± 0.19	3.15 ± 0.34	11.95 ± 1.37
Gamma-Terpinene	17.25	1130	1.57 ± 0.03	1.55 ± 0.04	1.62 ± 0.1	1.89 ± 0.02	1.53 ± 0.02	1.64 ± 0.06	1.6 ± 0.07	1.9 ± 0.18
Terpinolene	18.31	1290	0.04 ± 0	0.06 ± 0.02	0.43 ± 0.11	0.68 ± 0.03	0.23 ± 0.14	0.05 ± 0.01	0.44 ± 0.03	0.59 ± 0.08
cis Rose oxide	21.10	1356	1.56 ± 0.14	2.03 ± 0.26	2.69 ± 0.48	3.76 ± 0.15	1.54 ± 0.24	2.43 ± 0.22	2.11 ± 0.18	3.05 ± 0.22
Allo-Ocimene	21.52	1366	0.54 ± 0.03	0.81 ± 0.08	4.87 ± 0.59	4.66 ± 0.23	3.54 ± 0.2	3.14 ± 0.21	2.66 ± 0.09	2.94 ± 0.23
trans-Rose oxide	21.70	1370	0.73 ± 0.05	0.92 ± 0.11	1.04 ± 0.02	1.49 ± 0.05	0.71 ± 0.07	1 ± 0.06	0.97 ± 0.05	1.35 ± 0.09
Nerol oxide	26.05	1474	0.37 ± 0.01	0.54 ± 0.02	0.21 ± 0.04	0.06 ± 0	0.16 ± 0.11	0.36 ± 0.08	0.18 ± 0.08	0.42 ± 0.07
Linalool	29.05	1547	41.3 ± 0.52	49.38 ± 1.01	57.83 ± 0.49	274.24 ± 21.91	45.45 ± 3.55	48.21 ± 0.42	61.68 ± 1.96	227.02 ± 9.19
Hotrienol	31.64	1613	0.02 ± 0.01	0.03 ± 0.01	0.05 ± 0.01	0.07 ± 0	0.01 ± 0	0.04 ± 0	0.04 ± 0	0.03 ± 0
Terpinen-4-ol	31.78	1603	1.66 ± 0.14	2.84 ± 0.31	1.07 ± 0.03	1.49 ± 0.07	0.57 ± 0.02	1.12 ± 0.1	1 ± 0.07	2.57 ± 0.12
Neral	34.52	1689	0.25 ± 0.01	2.16 ± 0.24	0.08 ± 0	0.23 ± 0.01	0.1 ± 0.04	0.05 ± 0.02	0.16 ± 0.06	0.09 ± 0.02
a-Terpineol	35.02	1703	5.81 ± 0.08	2.48 ± 0.58	3.86 ± 0.04	6.01 ± 0.2	3.8 ± 0.05	5.38 ± 0.29	4.32 ± 0.09	8.26 ± 0.75
Citral	36.42	1741	34.89 ± 0.54	31.52 ± 0.02	38.33 ± 0.7	48.34 ± 0.82	33.19 ± 1	37.86 ± 1.01	36.97 ± 0.31	41.57 ± 0.76
cis-Pyranoid linalool oxide	36.45	1742	30.21 ± 0.44	40.45 ± 2.26	172.23 ± 187.06	454.71 ± 14.28	6.25 ± 0.3	24.52 ± 1.5	35.12 ± 1.07	247.13 ± 12
trans-Pyranoid linalool oxide	37.27	1765	31 ± 0.39	28.69 ± 2.12	10.75 ± 0.63	24.66 ± 1.15	9.52 ± 0.46	22.95 ± 1.43	16.06 ± 0.55	43.25 ± 4.3
Citronellol	37.47	1770	8.77 ± 0.12	6.91 ± 0.02	8.48 ± 0.14	8.52 ± 0.08	8.22 ± 0.05	8.78 ± 0.18	8.15 ± 0.06	10.53 ± 0.57
Nerol	38.81	1808	325.49 ± 3.98	578.88 ± 51.47	675.12 ± 21.87	1677.58 ± 120.83	394.2 ± 9.19	71.67 ± 2.88	1191.76 ± 84.46	1490.04 ± 40.86
Isogeraniol	39.10	1816	12.12 ± 0.16	3.17 ± 0.04	3.05 ± 0.49	5.06 ± 0.24	2.01 ± 0.05	2.77 ± 0.23	2.94 ± 0.14	19.67 ± 2.62
Geraniol	40.45	1855	127.05 ± 1.6	418 ± 0.99	698.77 ± 6.52	906.8 ± 43.84	144.32 ± 4.49	307.42 ± 30.61	403.71 ± 14.05	677.82 ± 18.92
Geranic acid	56.13	2350	14.07 ± 2.92	22.95 ± 1.17	19.27 ± 1.35	43.4 ± 2.07	7.99 ± 0.72	14.56 ± 2	15.85 ± 0.31	24.12 ± 2.87

## Data Availability

Data are contained within the article or Appendix A.

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
