# Peer review of "Integrated Transcriptomic and Metabolomic Analysis Revealed Abscisic Acid-Induced Regulation of Monoterpene Biosynthesis in Grape Berries"

_plants, 2024, doi:10.3390/plants13131862_

Round 1
Reviewer 1 Report
Comments and Suggestions for Authors
This manuscript ‘Integrated Transcriptomic and Metabolic Analysis revealed Abscisic Acid-induced Regulation of Monoterpene Biosynthesis in Grape Berries ‘ represents a good level of experimental research.
Main notes:
1. Perhaps, in the title of the article, the term metabolic should be replaced with metabolomic.
2. Abbreviation DEGs must be deciphered in Abstract.
3. Lines 26 - This expression is worth paraphrasing ‘enrichment in metabolic and secondary metabolic pathways,’
4. I propose replacing Table S2 and Table S3 from supplementary to the main text of manuscript and also adding the chromatograms of HS- SPME/GC‒MS analysis (to the supplementary).
5. Lines 567 - I suggest deciphering this abbreviation faster than in the conclusions ‘Weighted Gene Co-expression Network Analysis (WGCNA)’
Reviewer 2 Report
Comments and Suggestions for Authors
Review for
“Integrated Transcriptomic and Metabolic Analysis revealed Abscisic Acid-induced Regulation of Monoterpene Biosynthesis in Grape Berries”
The manuscript by Xiangyi Li and coworkers is a great fit for the special Plant issue on Flavor Quality of Cultivated and Wild Berries and Their Biological Basis. The authors first tested whether the application of abscisic acid (ABA solutions at 100 mg/mL and 500 mg/mL concentrations) onto grape berries resulted in significant changes of pH and total dissolved solids in comparison to control berries and then settled on the 500 mg/mL concentration for a detailed investigation on monoterpene content using HS-SPME/GC‒MS analysis (22 molecules were detected as listed in the supplement) followed by a transcriptome analysis to explain the biochemical basis for how ABA application regulates monoterpene biosynthesis. The transcriptome analysis resulted in the identification of several differentially expressed genes (DEGs). qRT-PCR analysis was used to validate six genes which were assigned to play a role in either the monoterpene or ABA biosynthesis pathway. A strength of the publication is that the authors were able to investigate grape berries at different ripening stages which they labeled EL-35, EL-36, EL-37, EL-38. Throughout the publication the authors present their data organized into the different ripening stages and it is interesting to see how monoterpene concentration levels change by ripening stage in addition to the ABA/control comparison (Figure 1) or how KEGG-pathways (Figure 5) and individual genes (Figure 7) shift in their expression levels according to the ripening stage and the ABA/control. Figure 7 is particularly useful for illustrating the biochemical basis of the ABA application effect, as it combines the pathway map for monoterpene and ABA biosynthesis with heatmaps for transcript abundance of genes along the pathways. Overall, the publication was very data rich and the results are interesting and well-explained. The authors made good use of bioinformatic and statistical tools, including different ways to represent data such as the radar plot in Figure 2 and a 3D PCA plot in Figure 3 (some publication still only show 2D PCA plots). There are, however, some minor points that need to be addressed as outlined in the following list.
1. The abstract contains abbreviations (HS-SPME/GC-MS in line 18 and DEGs in line 25) that need to be written out. The abstract also mentions a “dark green module” in line 33. The reader will not know what the dark green module is without reading the whole publication and finding the corresponding figure. I think it is figure 8 in which the color dark green is used to highlight a group of genes that are involved in grape maturation. Please ensure that the abstract can be understood without reading the complete manuscript.
2. The introduction makes it clear why a study that focuses on monoterpenes matters for the wine industry and gave a good overview on the described experiments and pathways involved in monoterpene biosynthesis. One gen, called TPS in line 79 was only listed as TPS. Please provide the full name.
3. Results: Please introduce the ripening stages early. Line 136 is the first time I encountered the phrase EL-37 and as a reader, I was unfamiliar with this abbreviation. The EL system is shortly mentioned in line 440 with a reference. Please provide a reference or even better a short explanation on this EL system earlier in your text.
4. Figure 3 contained very technical information and could be moved to the supplement.
5. Figure 6 could be enlarged as the gene names are difficult to read and Figure 8B looks a bit blurry (higher resolution please).
6. Some small mistakes:
a. Figure caption 8C refers to FPKM values but figures show RPKM as a label on the axis.
b. Line 474: What is a tampon solution?
c. Line 499 Illumine should be Illumina
d. a. Formatting of references needs to be checked
i. Reference 26 (name of author has capital letters)
ii. Reference 33 – I would not find this reference – what is COOMBEEBG?
iii. Reference 37 – all capital letters for authors
iv. Some references have a weird page break.
e. Supplemental Figure 1: y-axis reads “Total Soluble Solid” and needs to be “Total Soluble Solids”.
